# Triboelectric micromotors actuated by ultralow frequency mechanical stimuli

Hang Yang[1,2,5], Yaokun Pang[1,2,5], Tianzhao Bu[1,2], Wenbo Liu[1,2], Jianjun Luo[1,2], Dongdong Jiang[1,2], Chi Zhang ⓘ [1,2,3] & Zhong Lin Wang[1,2,3,4]

A high-speed micromotor is usually actuated by a power source with high voltage and frequency. Here we report a triboelectric micromotor by coupling a micromotor and a triboelectric nanogenerator, in which the micromotor can be actuated by ultralow-frequency mechanical stimuli. The performances of the triboelectric micromotor are exhibited at various structural parameters of the micromotor, as well as at different mechanical stimuli of the triboelectric nanogenerator. With a sliding range of 50 mm at 0.1 Hz, the micromotor can start to rotate and reach over 1000 r min$^{-1}$ at 0.8 Hz. The maximum operation efficiency of the triboelectric micromotor can reach 41%. Additionally, the micromotor is demonstrated in two scanning systems for information recognition. This work has realized a high-speed micromotor actuated by ultralow frequency mechanical stimuli without an external power supply, which has extended the application of triboelectric nanogenerator in micro/nano electromechanical systems, intelligent robots and autonomous driving.

[1] CAS Center for Excellence in Nanoscience, Beijing Key Laboratory of Micro-nano Energy and Sensor, Beijing Institute of Nanoenergy and Nanosystems, Chinese Academy of Sciences, 100083 Beijing, P.R. China. [2] School of Nanoscience and Technology, University of Chinese Academy of Sciences, 100049 Beijing, P.R. China. [3] Center on Nanoenergy Research, School of Physical Science and Technology, Guangxi University, 530004 Nanning, China. [4] School of Material Science and Engineering, Georgia Institute of Technology, Atlanta, GA 30332, USA. [5] These authors contributed equally: Hang Yang, Yaokun Pang. Correspondence and requests for materials should be addressed to C.Z. (email: czhang@binn.cas.cn) or to Z.L.W. (email: zlwang@gatech.edu)

The operation of a micro/nano electromechanical system (MEMS/NEMS) usually relies on an external power supply[1]. As the power consumption is decreasing, it is increasingly possible to develop a sustainable MEMS/NEMS supplied by the ambient environment energy[2–5]. An electrostatic micromotor, as a typical actuator in MEMS/NEMS with the use of electrostatic force and rotation momentum of inertia, has great advantages of simple structure, less energy consumption, and high speed[6,7], which has been widely used in the fields of Internet of Things, aerospace, and robots[8,9]. However, the required power supply with high voltage and frequency is a significant drawback for the high maintenance cost and finite lifetime[10–13], thus, a self-powered technology without external power supply is highly desired for a sustainable micromotor.

Recently, the triboelectric nanogenerator (TENG) was invented by Wang's group[14], which is based on the contact electrification and electrostatic induction. The TENG can effectively convert environmental mechanical energy and human body motion into electricity[15–17]. Various self-powered devices based on the TENG, such as photonic smart skin for tactile and gesture sensing, pressure and magnetic sensor, 3D acceleration sensor, have been developed[18–23]. Moreover, the TENG can easily produce high open-circuit voltage up to several thousand volts in low frequency, which has been used for tunable MEMS mirror, electrospinning system, microfluidic transport system, etc[24–28]. These researches have shown great potential of the TENG as a high-voltage source, which is very promising for actuating the micromotor.

In this work, a triboelectric micromotor (TM) is demonstrated by coupling a TENG with an electrostatic micromotor. Using the high output voltage of the TENG generated by the mechanical stimuli at an ultralow frequency, the micromotor can be activated and keeps a continuous rotation. The performances of the TM are exhibited at various structural parameters of the micromotor, as well as at different mechanical stimuli of the TENG. With a sliding range of 50 mm at 0.1 Hz, the micromotor can start to rotate and reach over 1000 r min$^{-1}$ at 0.8 Hz. By investigating the electric energy conversion, the operation efficiency of the TM is calculated at the maximum of 41%. Furthermore, the TM is demonstrated in two scanning systems for information recognition, in which the book ISBN can be recognized by slow hand motion and the moving obstacle can be detected by low-speed tire rolling, respectively. This work has realized a high-speed micromotor actuated by ultralow-frequency mechanical stimuli without an external power supply, which has shown the capability of TENG in driving MEMS/NEMS and has great significance and prospects toward independent, autonomous, and sustainable microsystems.

## Results

**Mechanism of triboelectric micromotor.** A schematic illustration of the TM is shown in Fig. 1a. The TM is composed of a TENG in free-standing mode, a rectifier and a micromotor, in which the TENG can generate a high driving voltage through the rectifier for the micromotor, by ultralow mechanical stimuli. The micromotor consists of a rotor (1 cm in radius), a shaft, and two vertical electrodes with carbon fibers stuck on. The micromotor is fabricated with an accurately controlled size by 3D printing, which has four blades coated with copper films. The detailed fabrication process has been revealed in Supplementary Fig. 1. Two dissimilar materials, copper, and polytetrafluoroethylene (PTFE) are selected as the frictional layers for the TENG. The inset is a scanning electron microscopy (SEM) image of the PTFE.

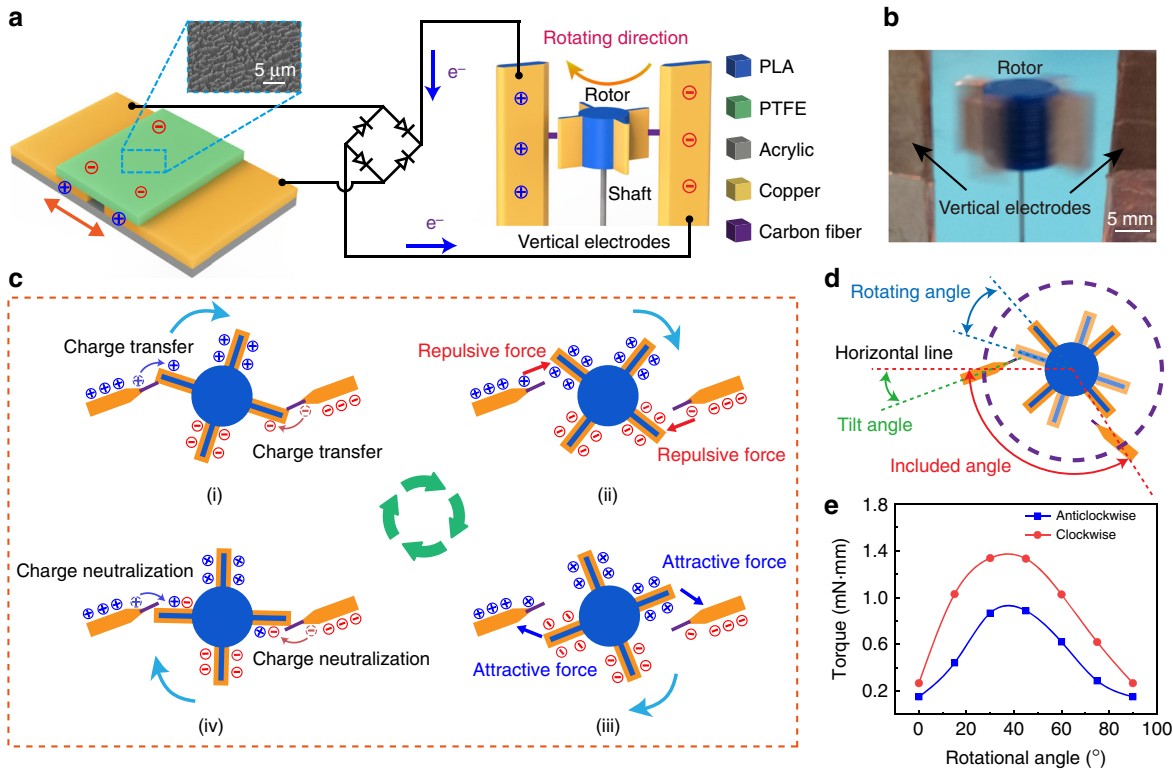

**Fig. 1** Schematic illustration and working principle of the triboelectric micromotor. **a** Schematic diagram of the TM and SEM image of the PTFE film. **b** Image of the rotating micromotor. **c** Working process of the micromotor during a quarter of cycle by the Coulomb forces of accumulated charges. **d** Diagrammatic sketch of the rotating angle, tilt angle and included angle. **e** Simulated curves between rotor torque and rotation angles during a quarter of cycle clockwise and anticlockwise with four blades and included angle of 180 degrees

Using the inductively coupled plasma (ICP), the surface of PTFE film was etched with nanostructures, which can increase the effective surface area and improve the triboelectric charge density. In Supplementary Fig. 2a, when the PTFE film contacts with the copper foil for electrification, the PTFE film has negative charges while the copper foil has positive charges due to the different charge affinities. When the PTFE film moves by the external mechanical stimuli, some of the positive charges on the copper foil will lose restraints and tend to transfer from the left one to the right one, which makes charges accumulate on the two vertical electrodes through the rectifier. With the PTFE film moving back and forth, the charges could be continuously accumulated to activate the micromotor by the Coulomb force as a high-voltage power supply. In this process, the high voltage between the two vertical electrodes can reach the starting voltage of the micromotor, which is about 1800 V if powered by a general direct current source. The operation process of the TM is shown in Supplementary Movie 1. Figure 1b is the image of the rotating micromotor, in which the rotor and vertical electrodes are marked.

The working process of the micromotor is presented in Fig. 1c. The carbon fibers, stuck on the vertical electrodes, contact with the blades in the initial state as shown in Fig. 1c (i). The left-hand vertical electrode is electrified with positive charges while the right one with negative charges. Due to the conductivity of carbon fibers, charges on the two vertical electrodes transfer to the two rotor blades, respectively. According to the Coulomb's law, the charged blades separate from the vertical electrodes by the repulsive force between the same sign of charges, and the micromotor starts to rotate clockwise, as shown in Fig. 1c (ii). With the rotation momentum of inertia, the two blades gradually rotate away from the previous vertical electrode and approach to the other one, respectively. The repulsive forces gradually decrease and the attractive forces start to dominate for the opposite charges between them, as shown in Fig. 1c (iii). As the blades contact with the carbon fibers in Fig. 1c (iv), the previous charges are neutralized and the two blades are oppositely charged, respectively. After that, the micromotor comes to the initial state again. The micromotor can keep rotating by a strong electric field, with the continuously accumulated charges from the TENG to the vertical electrodes. This is the basic operation mechanism of the TM. From the perspective of energy conversion, this process can be illustrated in Supplementary Fig. 2b that the electric energy generated by TENG can be converted into the rotary motion of micromotor without external power supply. It is worth noting that if the initial state is selected as shown in Fig. 1c (ii), the micromotor is also able to start up, just harder than the state in Fig. 1c (i). By polarizing air, the blades on the rotor can still be electrified by charges transferred from the vertical electrodes. Once the blades are sufficiently charged, the TM can start to work.

To explain the rotating direction of the rotor, the electric potential simulation of the micromotor is carried out by the finite element method in COMSOL software[29]. The electric potential distribution (depicted in Supplementary Fig. 2c) is calculated at different rotation angles. Generally, there are two possible rotating directions in clockwise and anticlockwise. In order to determine the rotating direction, a tilt angle between the electrodes and horizontal line is set, which is shown in Fig. 1d. From the electric potential distribution, a simulated torque of the rotor is calculated at different rotation angle, depicted in Fig. 1e, in which two simulated curves are displayed during a quarter of cycle. Seeing that the rotor has four blades, the simulated torque curves are going to repeat for every quarter of cycle. The red curve is the torque of the rotor when it turns clockwise, while the blue one is in anticlockwise, indicating that the micromotor tends to turn clockwise rather than anticlockwise, in virtue of the larger torque in this direction. Also, the relationship between the torque and rotating angle with different tilt angles is calculated in Supplementary Fig. 3a, which indicates the differences in tilt angles can be ignored. In this work, the tilt angle is set to 20 degrees.

**Performances of triboelectric micromotor.** With different structure parameters and mechanical stimuli, the performances of TM are systematically simulated and experimentally tested. Firstly, the performances dependences on structure parameters are investigated. As experimental measurement in Fig. 2a, with the included angle (defined in Fig. 1d) of 180 degrees and sliding range of 100 mm at 0.8 Hz, the rotation rate of the micromotor goes up as the blade number increases at the beginning, for more charges can be easily transferred and a larger drive torque is produced, as simulated in Supplementary Fig. 3c. However, more blades create larger air resistance and friction moments, as simulated in Supplementary Fig. 4, which has reduced the rotation rate. When the blade number is 7, the maximum rotation rate can reach 1300 r min$^{-1}$, which is consistent with the simulated variation trend in Supplementary Fig. 5b. Besides, the rotation rate of the micromotor with different included angles is shown in Fig. 2b, at the blade number of 7. With the increase of the included angle, the rotation rate of the micromotor rises. When the included angle is set at 180 degrees, the maximum value of the rotation could be reached. It is because that the blades moving between the two electrodes is the accelerating process of the rotor. As the included angle increases, the two electrodes get farther, which has increased the accelerating distance and rotation rate. As well, Supplementary Fig. 3b has shown the simulated average electrostatic torque versus included angle at different blade number, in which the Column force can generate the maximum torque for the rotor with the included angle of 180 degrees. The simulated variation trend in Supplementary Fig. 5a has also predicted the actual behavior in Fig. 2b. In order to evaluate the load capacity of the micromotor, extra loads are put on the rotor with seven blades and included angle of 180 degrees. As depicted in Fig. 2c, the rotation rate of the micromotor decreases with the weight in a linear relation, for the greater weights create larger friction resistance between the rotor and shaft. Nevertheless, the ability of the micromotor for carrying loads is commendable for the maximum load is 5.21 g, which is over five times of the rotor weight (1.01 g in mass).

To figure out the effects of mechanical stimuli on TM, we investigate different sliding ranges and frequencies of the TENG with the optimal structure parameters of the micromotor. As presented in Fig. 2d, with the increases of the sliding range at the frequency of 0.8 Hz, the rotation rate of the micromotor rises. When the sliding range is up to 80 mm, the rotation rate of the micromotor seems to remain at around 1300 r min$^{-1}$ and the growth rate comes to decline. It is noted that when the sliding range is less than 30 mm, the output voltage cannot achieve the starting voltage, thus the micromotor cannot be activated. The frequency of TENG has the similar variation tendency on the rotation rate. As shown in Fig. 2e, when the sliding range is 100 mm, the motor can start to rotate with 880 r min$^{-1}$ at an ultralow frequency of 0.1 Hz, and comes to 1350 r min$^{-1}$ at 0.8 Hz. With the frequency around 0.8 Hz, the growth rate falls and the rotation rate tends to reach the saturation. The rotation rate with different frequencies and sliding ranges are exhibited in Supplementary Fig. 6. To illustrate this phenomenon, the rotation rate versus accumulated charge quantity with different blade numbers and included angles are simulated in Supplementary Fig. 5c, d. The increases of transferred charges with larger

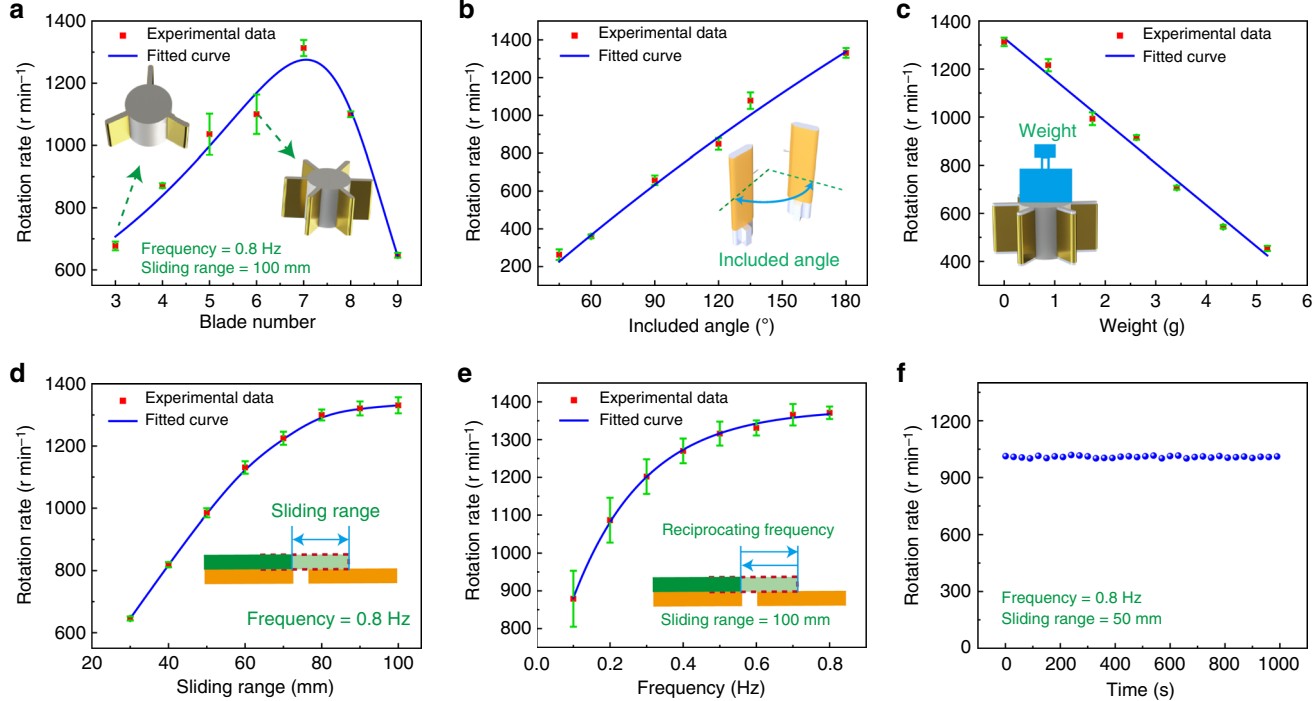

**Fig. 2** Triboelectric micromotor performances with different parameters and stimuli. **a–c** Dependence of rotation rate on blade number, included angle, and load weight, respectively, with sliding range of 100 mm at 0.8 Hz. **d, e** Dependence of rotation rate on sliding range and frequency, respectively, with optimized structure parameters. **f** The stability test of the optimized TM with the sliding range of 50 mm at 0.8 Hz. All error bars in the figure represent s.d. of the data

range can lead to larger torque as well as the rotation rate of the micromotor. Also, when the frequency increases, more charges will be transferred in the same way. However, when the transferred charges on the blades reach to a certain amount, the blades are separated from the vertical electrodes by the repulsive force. Thus, the charges cannot be indefinitely accumulated and will reach a limitation. The simulated air resistance and friction moments in Supplementary Fig. 4 indicate the increase of motor rotation rate will cause much larger resistant moment. With the action of limited charges and resistant moments, the rotation rate of the micromotor tends to reach the saturation. The rotating characteristics simulations of the TM are elaborated in Supplementary Note 1. In addition, the stability test of the micromotor at the rotation rate of 1000 r min$^{-1}$ with the sliding range of 50 mm and the frequency of 0.8 Hz is presented in Fig. 2f, as well as the stability test of the sliding mode TENG in Supplementary Fig. 12, which indicate the TM has a good stability of long-running for further application.

**Operation efficiency of triboelectric micromotor.** The output electric energy $E_e$ from the TENG to the micromotor, in one cycle, can be described as the encircled area of the closed loop in the U–Q curve[30,31]:

$$E_e = \oint U_{TM} dQ_{TM} \tag{1}$$

where $U_{TM}$ and $Q_{TM}$ are the output voltage and transferred charge from the TENG to the micromotor.

As the rotational kinetic energy $E_r$ can be described as:

$$E_r = \frac{1}{2} J \omega^2 = \frac{1}{2} J \left( \frac{2\pi n}{60} \right)^2 \tag{2}$$

where $J$ is the rotational inertia of the rotor, which was calculated by the finite element method, $\omega$ is the angular velocity of the

rotor, and $n$ is the rotation rate, the energy conversion efficiency of the TM $\eta$ is defined as:

$$\eta = \frac{E_r}{E_e} \tag{3}$$

which is the ratio between the rotational energy and electric energy.

To estimate the energy conversion efficiency of the TM, the characteristics are illustrated with the optimal structure parameters. Supplementary Fig. 7a, b shows the measured voltage and charge waveforms of the TENG and Fig. 3a summarizes the measured peak-to-peak values with the sliding range of 30 mm, whereas the frequency varied from 0.3 to 0.8 Hz. As depicted, the voltage rises from 880 V to 1050 V in a degressive growth and the transferred charge keeps a stable value about 280 nC with the increases of the frequency. In comparison with the open-circuit voltage and short-circuit charge of the TENG, as shown in Supplementary Fig. 9a, the voltage and transferred charge have both decreased for the charge consumption in the micromotor. The U–Q curves of the TM in one cycle with different frequencies at the sliding range of 30 mm are plotted in Fig. 3b, which tends to overlap with the frequency exceeding 0.6 Hz. The rotational energy, electric energy, and operation efficiency are calculated and summarized in Fig. 3c. The electric energy $E_e$ gradually increases from 0.15 mJ and tends to keep a stable value of 0.19 mJ at a higher frequency. Also, the rotational kinetic energy $E_r$ starts at 41 μJ and inclines to reach saturation of 77 μJ at a higher frequency for the rotation rate tends to become saturated. Consequently, the energy conversion efficiency $\eta$ with the frequency exhibits a gradual growth with slight increase from 27% and tends to reach saturation of 41% at a higher frequency.

In addition, the sliding range of the TENG as an important influence factor to the operation efficiency has also been systematically investigated. The electric characteristics from the TENG to the micromotor are measured with different sliding

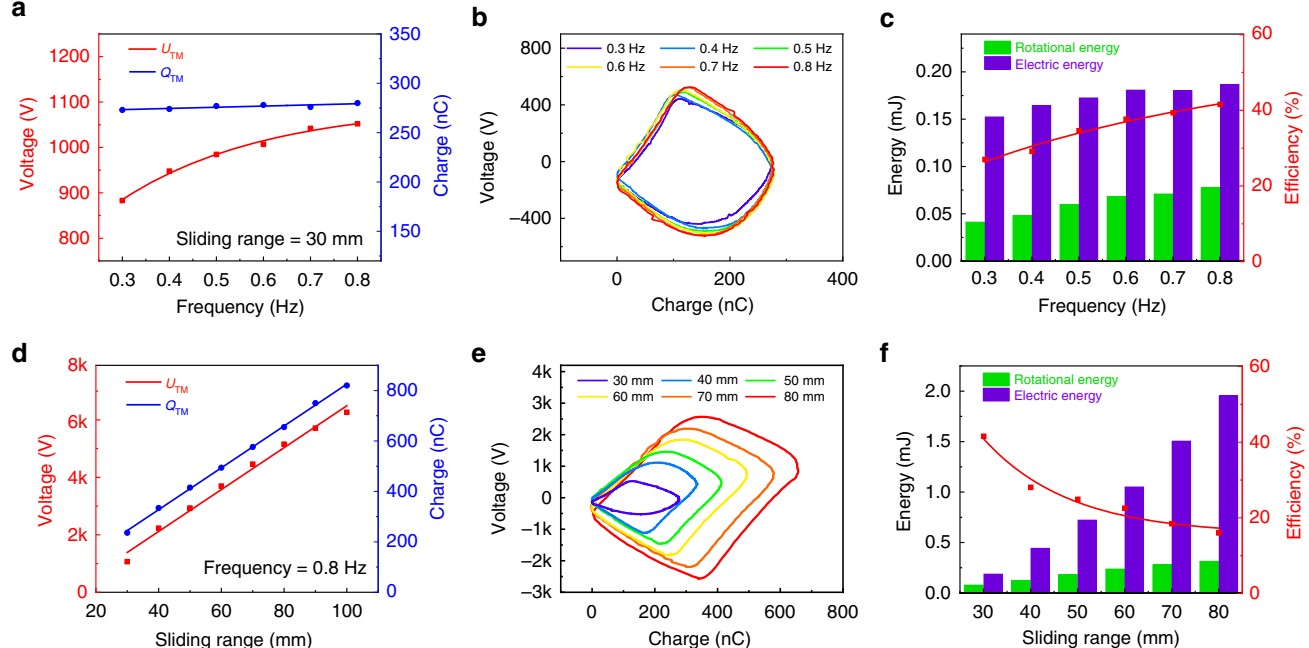

**Fig. 3** Output of the nanogenerator and operation efficiency of the micromotor. **a**, **b** Measured peak-to-peak voltages, transferred charges and U–Q plots of TM with different frequencies at the sliding range of 30 mm. **c** Rotational energy, electric energy and operation efficiency of the TM with different frequencies at the sliding range of 30 mm. **d**, **e** Measured peak-to-peak voltages, transferred charges and U–Q plots of TM with different sliding ranges at 0.8 Hz. **f** Rotational energy, electric energy and operation efficiency of the TM with different sliding ranges at 0.8 Hz

ranges from 30 mm to 100 mm at a frequency of 0.8 Hz, as shown in Supplementary Fig. 7c, d. The measured peak-to-peak values are summarized in Fig. 3d. With the increases of the sliding range, the output voltage and transferred charge increase, respectively, and they all have shown a good linear relationship to the sliding range. Similarly, the voltage and transferred charge are both less than the open-circuit voltage and short-circuit charge of the TENG in Supplementary Fig. 9b for the charge consumption. The U–Q curves of the TENG with micromotor at different sliding ranges at 0.8 Hz are plotted in Fig. 3e. The experimental results are consistent with the simulated curves and variation trends in Supplementary Fig. 10b, based on an equivalent circuit model of the TM in Supplementary Fig. 10a and detailed theoretical derivation in Supplementary Note 2. The rotational energy, electric energy and operation efficiency are also calculated and summarized in Fig. 3f. Obviously, the electric energy $E_e$ increases from 0.19 mJ to 1.95 mJ for both $U_{TM}$ and $Q_{TM}$ grow linearly with the sliding range. However, the rotational kinetic energy $E_r$ increases from 77 μJ and then reach the saturation of 313 μJ with the similar tendency of rotation rate as shown in Fig. 2d. Thus, the energy conversion efficiency $\eta$ decreases with the increasing sliding range. It is worth noting that the maximal $\eta$ of 41% can be reached with the sliding range of 30 mm at the frequency of 0.8 Hz. The electric energy loss could be friction dissipation, air resistance dissipation, and storage in the capacitor of the two vertical electrodes, as shown in Supplementary Fig. 2b.

**Triboelectric micromotor for information recognition**. Furthermore, we demonstrate the TM for optical scanning based on the advantages of high speed and tiny size. The schematic diagram of optical scanning[32,33] is shown in Fig. 4a, in which a structure of hexagonal mirror is fabricated on the top of the micromotor. With the rotation of the micromotor, an external laser beam hits on the mirror and forms a scanning ray as shown in Fig. 4b. The scanning ray received by screen is recorded using a high-speed camera. The relationship between spot position and

angle of incidence is drawn in Fig. 4c, which indicates rotation stationarity of the micromotor. Besides, the inset is images of spot track at different moments, which is approximately even distribution.

On the basis, two scanning systems for information recognition are demonstrated, respectively. Figure 4d depicts a TM-based portable scanner, in which the multilayered mechanical sliding generated by hand motion at an ultralow frequency of about 0.8 Hz has been converted into electricity and used to drive the micromotor for producing the scanning ray. When the scanning line falls on the ISBN, the barcode reader is activated and recognizes the codes (Supplementary Movie 2). Moreover, a TM-based moving obstacle detector is presented in Fig. 4e. The rotational sliding TENG with tire rolling could be used for actuating the micromotor and scanning the moving obstacle. Even though the tire rotates at an ultralow speed of 11.3 r min⁻¹, the high-speed scanning could also be realized without external power supply (Supplementary Movie 3), which has great application prospects in vehicle-mounted laser radar and automatic drive. These results have demonstrated the high-speed micromotor is driven by the wasted ultralow-frequency energy in human and ambient, with great significance and prospects toward the autonomous information recognition technology.

## Discussion

In summary, we reported a TM by coupling a micromotor and a TENG. Using the high output voltage generated by the TENG at ultralow-frequency mechanical stimuli, a strong electric field was created between the two vertical electrodes for actuating the micromotor. Through periodic charge transfer and neutralization, the micromotor could maintain continuous operation by the electrostatic force and rotation momentum of inertia. The performances of the TM were investigated in detail at different structural parameters of the micromotor, in which the optimal blade number of the micromotor was 7 and the included angle

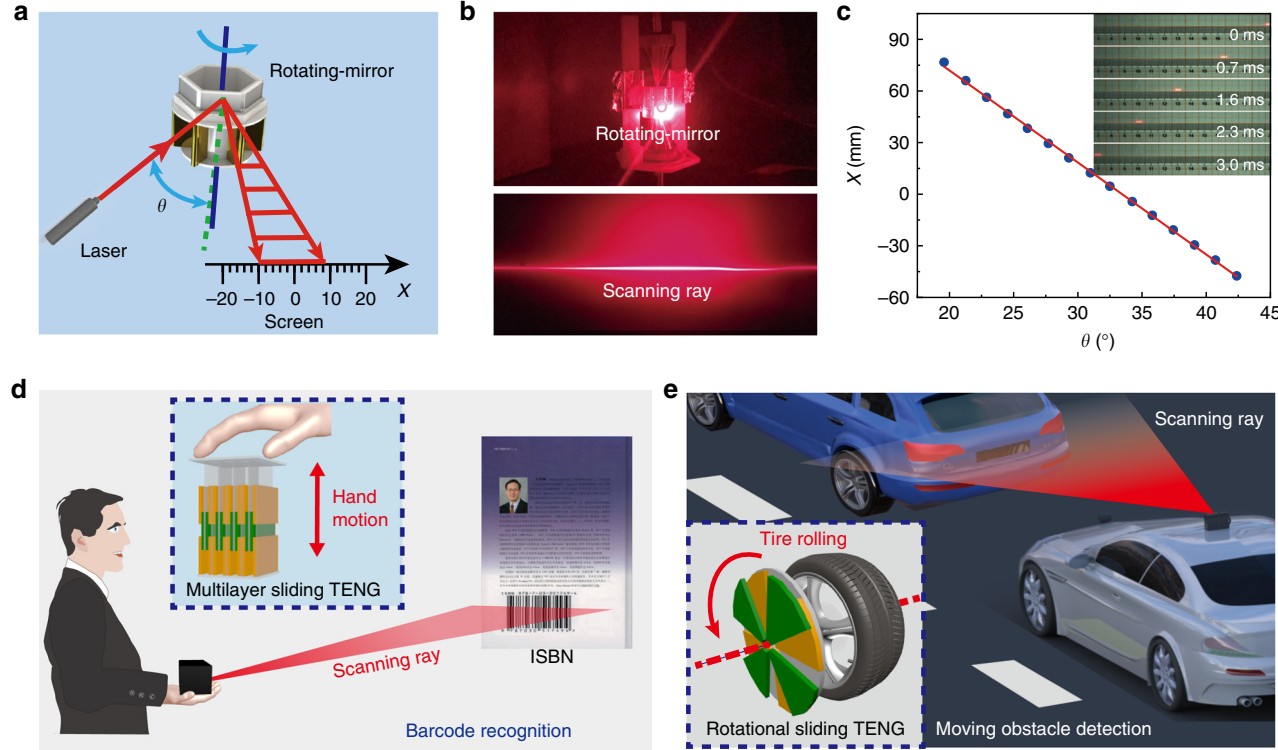

**Fig. 4** Triboelectric micromotor for optical scanning and identity recognition. **a** Schematic diagram of the TM for optical scanning. **b** Images of the micromotor and scanning ray. **c** Relationship between spot position and angle of incidence. Inset is spot positions at different moments. **d** Schematic illustration of the TM-based portable scanner for book ISBN recognition by slow hand motion. **e** Schematic illustration of the TM-based scanning system for moving obstacle detection by low-speed tire rolling

was 180 degrees. At an ultralow frequency of 0.1 Hz with the sliding range of 50 mm, the rotation rate of the micromotor can start to rotate and reach over 1000 r min$^{-1}$ at 0.8 Hz. Also, the electric energy for maintaining the stable rotation of the micromotor was calculated and the operation efficiency of the TM reached a maximum value of 41%. Furthermore, the TM was demonstrated in two scanning systems for information recognition by the ultralow-frequency energy in human and ambient, in which the book ISBN can be recognized by slow hand motion and the moving obstacle can be detected by low-speed tire rolling, respectively. This study provides a way of actuating a high-speed motor by ultralow-frequency mechanical stimuli without an external power supply and expands the application of TENG in MEMS/NENS, intelligent robots and autonomous driving.

## Methods

**Fabrication of the triboelectric nanogenerator**. The free-standing mode TENG is consisted of two frictional components. One is a fixed base, and the other is the sliding part. Acrylic substrate which is 5 mm in thickness and 185 × 185 mm in size acted as the base. Two copper foils were adhered to the fixed base, which were the same in size and had the thickness of 150 μm, acting as the electrodes. It's noted that there was a 5 mm wide gap between the two foils. As for the sliding part, a sponge which is 8 mm in thickness was attached to the acrylic substrate in order to get better surface contact. Then the etched PTFE film which is 200 μm in thickness was stuck on the sponge.

**Fabrication of the electrostatic micromotor**. All the parts of the micromotor including the rotor and the electrodes were designed in Solidworks software. After that, the designed models were made by 3D printing technology. The 3D printing order was received by WeNext Factory (Alison.Li) whose machining accuracy is 0.1 mm. The rotor with a 1-mm-wide blade is 10 mm in radius and 10 mm in height, together with the vertical electrodes were coated with copper foils. After that, carbon fibers were fixed on the vertical electrodes. Finally, a shaft with diameter of 0.7 mm is inserted into a cylindrical hole with diameter of 1.2 mm in the center of the rotor, and assembled with the vertical electrodes. The manufacturing process is shown in Supplementary Fig. 1.

**Electric simulation**. The calculation of 2D potential distribution between two electrodes of the micromotor and the torque on the rotor were carried out in the commercial COMSOL software. The floating potential with charge magnitude of 1E-6 C is assigned to the electrodes, and 1E-9 C is assigned to the blades in the simulation. The sizes were consistent with those of the designed 3D models.

**Characterization**. The transferred charges were measured by an electrometer (Keithley, 6514), and the voltages were measured by an electrostatic probe (Trek-347). During the measurement, one end of the measured device was grounded, while the other end was attached to a Copper film which was 5 mm beneath the probe. The diagrammatic illustration is shown in Supplementary Fig. 13. The rotation rate of the micromotor was measured by the velocimeter (UT372) with measurement range of 0–9999 r min$^{-1}$.

## Data availability

All data needed to evaluate the conclusions in the paper are present in the paper and/or the Supplementary Information. Additional data related to this paper may be requested from the authors. The source data underlying all figures can be found in the Source Data file.

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

## Acknowledgements

This work is supported by the National Natural Science Foundation of China (Nos. 51475099, 61874011), National Key Research and Development Program of China (2016YFA0202704), Beijing Talents Foundation (2017000021223TD04), Beijing Nova Program (No. Z171100001117054). Patent has been filed. We also thank Dr. Tao Jiang, Dr. Zhi Zhang, and Dr. Guoxu Liu for the assistance in simulation and demonstration.

## Author contribution

H.Y., C.Z., and Z.L.W. conceived the idea, H.Y., Y.K.P., C.Z., and Z.L.W. analyzed the data and wrote the paper. H.Y., Y.K.P., and C.Z. designed the structures of the tribo-electric micromotor. W.B.L. helped with the COMSOL simulations. T.Z.B. and D.D.J. helped with the experiments. J.J.L. helped with the high-speed photography. All the authors discussed the results and commented on the manuscript.

## Additional information

**Competing interests:** The authors declare no competing interests.

