## [Peer Review File · Nature Communications]

Reviewers' Comments:

Reviewer #1:

Remarks to the Author:

This paper draft describes the development of a high-speed micromotor that couples a micromotor and triboelectric nanogenerator. The type of system described is novel to the reviewer's knowledge. Hence, the content should influence the thinking in this field. The system described by the authors sounds exciting to the reviewer.

The reviewer has a few concerns that the authors may consider.

1. The correlation of the COMSOL model with the actual behavior of the device could be more developed. Though the results from the model are described in detail, the relationship between the model predictions and the actual device are not in a meaningful way.
2. The description of the method used for the device fabrication could be more detailed.
3. Other forms of regenerative or low power consumption devices have been reported in the literature, though these devices are not identical to the one described in this paper. The paper quality may be enhanced if this device were compared to one or two other devices of similar application in terms of power density and efficiency.

It would have also been interesting if the authors had described more meaningfully the differences in the voltage versus sliding range and voltage versus charge loops depicted in Figure 3.d and 3.e, respectively. Why is there overlap in the plots in Figure 3.d and is there a relationship between the voltage hysteresis and the sliding range that can be described mathematically based on the ranges studied?

Reviewer #2:

Remarks to the Author:

The author reported a high-speed triboelectric motor driven by sliding mode TENG. The author analyzed the effect of various parameters on the TM. The initial analysis of TEM by the author is very interesting but the application part (identity recognition) is too vague and impractical. I would like to suggest a major revision.

1. The author is advised to provide the output current at different frequencies.
2. The author measured the output using Trek 347. The range of Trek 347 is 3KV but some values of output are higher than 3KV. How the author measured that high voltage.
3. The author showed open circuit voltage in the supplementary figure. The trek 347 is surface voltage measurement. The author should mention in this case how voltage varies with the load resistance.
4. The author is advised to report the stability of sliding mode TENG to confirmed long-term applicability.
5. The application shown by the author is simply a replication of bar code reader where scanning ray created by TM. What is the advantage of such a system over commercial barcode reader? The author should demonstrate more appropriate application for the use of TM. Furthermore, the author claimed it a self-powered system but there are so many components involves which needs external power. The self-powered word is too weak in the context.
6. The author demonstrated a human identity recognition system. The reported system is very impractical for real application. The device shown in the figure looks like a contact-separation device but the author used a sliding mode device for other analysis.

Reviewer #3:
None

Revisions and Responds

Manuscript ID: NCOMMS-18-30792

Manuscript Type: Article

Title: Triboelectric micromotors actuated by ultralow frequency mechanical stimuli

Author(s): Hang Yang, Yaokun Pang, Tianzhao Bu, Wenbo Liu, Jianjun Luo, Dongdong Jiang, Chi Zhang, Zhong Lin Wang

Corresponding author: Chi Zhang, czhang@binn.cas.cn; Zhong Lin Wang, zlwang@gatech.edu

Dear editor and reviewers,

Thank you so much for giving us an opportunity to revise this manuscript, and we are quite appreciative for your comments and valuable suggestions. We have done a lot of work to address the concerns and this manuscript has been greatly improved. In particular, according to the comments of Reviewer #1, we have mainly carried out more simulations and theoretical analysis on the TM, which have predicted the actual device behavior. Also, the comparison with similar devices has been included. For the comments of Reviewer #2, the concerns about voltage and stability have been addressed, and we have made major revisions in the application part. Appended are the revisions that we made to the manuscript as well as the responds to each of the reviewer's comments. In the marked up revised manuscript, all the revisions are highlighted. We wish the revisions could address all the concerns and receive your supports. Thank you very much.

List of revisions made to the manuscript and supplementary information:

Rev 1. Page 1, line 28-30: Additionally, the TM is demonstrated in two scanning systems for information recognition, in which the book ISBN can be recognized by slow hand motion and the moving obstacle can be detected by low-speed tire rolling, respectively.

Rev 2. Page 2, line 30-33: Furthermore, the TM is demonstrated in two scanning systems for information recognition, in which the book ISBN can be recognized by slow hand motion and the moving obstacle can be detected by low-speed tire rolling, respectively.

Rev 3. Page 3, line 2-3: The detailed fabrication process has been revealed in Supplementary Fig. 1.

Rev 4. Page 4, line 17-27: With different structure parameters and mechanical stimuli, the performances of TM are systematically simulated and experimentally tested. Firstly, the performances dependences on structure parameters are investigated. As

experimental measurement in Fig. 2a, with the included angle (defined in Fig. 1d) of 180 degrees and sliding range of 100 mm at 0.8 Hz, the rotation rate of the micromotor goes up as the blade number increases at the beginning, for more charges can be easily transferred and a larger drive torque is produced, as simulated in Supplementary Fig. 3c. However, more blades create larger air resistance and friction moments, as simulated in Supplementary Fig. 4, which has reduced the rotation rate. When the blade number is 7, the maximum rotation rate can reach 1300 r/min, which is consistent with the simulated variation trend in Supplementary Fig. 5b.

Rev 5. Page 4, line 34-38: As well, Supplementary Fig. 3b has shown the simulated average electrostatic torque versus included angle at different blade number, in which the Column force can generate the maximum torque for the rotor with the included angle of 180 degrees. The simulated variation trend in Supplementary Fig. 5a has also predicted the actual behavior in Fig. 2b.

Rev 6. Page 5, line 13-15: To illustrate this phenomenon, the rotation rate versus accumulated charge quantity with different blade numbers and included angles are simulated in Supplementary Fig. 5c and 5d.

Rev 7. Page 5, line 21-23: The simulated air resistance and friction moments in Supplementary Fig. 4 indicate the increase of motor rotation rate will cause much larger resistant moment.

Rev 8. Page 5, line 24-26: The rotating characteristics simulations of the TM are elaborated in Supplementary Note 1.

Rev 9. Page 6, line 3-6: Supplementary Fig. 7a and 7b show the measured voltage and charge waveforms of the TENG and Fig. 3a summarizes the measured peak-to-peak values with the sliding range of 30 mm, whereas the frequency varied from 0.3 to 0.8 Hz.

Rev 10. Page 6, line 22-25: The electric characteristics from the TENG to the micromotor are measured with different sliding ranges from 30 mm to 100 mm at a frequency of 0.8 Hz, as shown in Supplementary Fig. 7c and 7d. The measured peak-to-peak values are summarized in Fig. 3d.

Rev 11. Page 6, line 31-34: The experimental results are consistent with the simulated curves and variation trends in Supplementary Fig. 10b, based on an equivalent circuit model of the TM in Supplementary Fig. 10a and detailed theoretical derivation in Supplementary Note 2.

Rev 12. Page 7, line 13-24: On the basis, two scanning systems for information recognition are demonstrated, respectively. Fig. 4d depicts a TM-based portable

scanner, in which the multilayered mechanical sliding generated by hand motion at an ultralow frequency of about 0.8 Hz has been converted into electricity and used to drive the micromotor for producing the scanning ray. When the scanning line falls on the ISBN, the barcode reader is activated and recognizes the codes (Supplementary Video 2). Moreover, a TM-based moving obstacle detector is presented in Fig. 4e. The rotational sliding TENG with tire rolling could be used for actuating the micromotor and scanning the moving obstacle. Even though the tire rotates at an ultralow speed of 11.3 r/min, the high-speed scanning could also be realized without external power supply (Supplementary Video 3), which has great application prospects in vehicle-mounted laser radar and automatic drive.

Rev 13. Page 7, line 41-42; Page 8, line 1-2: Furthermore, the TM was demonstrated in two scanning systems for information recognition by the ultralow-frequency energy in human and ambient, in which the book ISBN can be recognized by slow hand motion and the moving obstacle can be detected by low-speed tire rolling, respectively.

Rev 14. Page 8, line 23-26: Finally, a shaft with diameter of 0.7 mm is inserted into a cylindrical hole with diameter of 1.2 mm in the center of the rotor, and assembled with the vertical electrodes. The manufacturing process is shown in Supplementary Fig. 1.

Rev 15. Page 13, line 1: Fig. 3d.

Rev 16. Page 14, line 1-7: Fig. 4.

Rev 17. Supporting Information, Page 2, line 1-2: Supplementary Figure 1.

Rev 18. Supporting Information, Page 3, line 1-6: Supplementary Figure 3.

Rev 19. Supporting Information, Page 3, line 8-12: Supplementary Figure 4.

Rev 20. Supporting Information, Page 4, line 1-5: Supplementary Figure 5.

Rev 21. Supporting Information, Page 5, line 1-5: Supplementary Figure 7.

Rev 22. Supporting Information, Page 5, line 7-8; Page 6, line 1-3: Supplementary Figure 8.

Rev 23. Supporting Information, Page 6, line 5-10: Supplementary Figure 9.

Rev 24. Supporting Information, Page 6, line 12-14: Supplementary Figure 10.

Rev 25. Supporting Information, Page 7, line 1-3: Supplementary Figure 11.

Rev 26. Supporting Information, Page 7, line 5-6: Supplementary Figure 12.

Rev 27. Supporting Information, Page 8, line 1: Supplementary Table 1

Rev 28. Supporting Information, Page 8, line 3-29; Page 9, line 1-7: Supplementary Note 1.

Rev 29. Supporting Information, Page 9, line 9-30: Supplementary Note 2.

Rev 30. Supporting Information, Page 10, line 2-12: Supplementary Note 3.

Rev 31. Supporting Information, Page 10, line 14-28: Supplementary References.

Rev 32. Supplementary Video 2.

Rev 33. Supplementary Video 3.

Respond to the reviewers:

Reviewer #1: *This paper draft describes the development of a high-speed micromotor that couples a micromotor and triboelectric nanogenerator. The type of system described is novel to the reviewer's knowledge. Hence, the content should influence the thinking in this field. The system described by the authors sounds exciting to the reviewer.*

Respond: Thank you very much for your positive comments on our work. We will make persistent efforts.

Reviewer #1: *The correlation of the COMSOL model with the actual behavior of the device could be more developed. Though the results from the model are described in detail, the relationship between the model predictions and the actual device are not in a meaningful way.*

Respond: Thank you very much for your suggestions. We have added more simulation and calculation on the rotating characteristics of TM, which is very helpful for the behavior predictions of actual devices. All the simulation results in this work are summarized as below.

Firstly, we have simulated the driving torque of rotor with a tilt angle of 20 degrees in Fig. 1e, indicating that the micromotor tends to turn clockwise rather than anticlockwise, in virtue of the larger torque in this direction. As well, the driving torque curves with different tilt angles almost coincide in Supplementary Fig. 3a, which indicates that the tilt angle can only determine the rotational direction but not the driving torque of the micromotor.

Secondly, the simulations on the resistant moment of rotor are elaborated in Supplementary Note 1 and Supplementary Fig. 4, including air resistant moment and friction moment. The air resistant moment is very small and can be ignored, while the friction moment is dominant and goes up with the increasing rotation rate and blade number, in virtue of the increasing rotational centripetal force and friction force on the shaft. Therefore, the total resistant moment is determined by the both rotation rate and blade number. Moreover, the driving torque increases with the blade number for more

transferred charges, which is simulated in Supplementary Fig. 3c. For the balanced state that the driving and resistant moments are equal, the rotation rate can be calculated with different blade number in Supplementary Fig. 5b. It is demonstrated that the optimal blade number is 7, which has exactly predicted the measured variation trend in Fig. 2a.

Thirdly, the driving torque grows up with the increase of included angle till 180 degrees, which is simulated in Supplementary Fig. 3b. As the resistant moment is not determined by the included angle, the larger included angle can give rise to the larger rotation rate, which is shown in Supplementary Fig. 5a and consistent with the measured variation tendency shown in Fig. 2b.

Finally, we have investigated the impact of charge quantity on the rotation rate in Supplementary Fig. 5c and 5d, which show that the micromotor rotates faster with more charges. With the increase of either sliding range or frequency, the TENG can produce more charges in one cycle. Therefore, the rotation rate rises with the increasing sliding range and frequency, which are consistent with the measured variation trends in Fig. 2d and 2e.

To sum up, the simulation results have predicted the actual device behavior and the experimental results have verified the structure model, which are beneficial to the device design, optimization and practical application. The relevant revisions are made in **Rev 4-8, 18-20, and 28**.

Reviewer #1: *The description of the method used for the device fabrication could be more detailed.*

Respond: Thank you very much for your reminder. We have added the description of the device fabrication method and the detailed process is shown in Supplementary Fig. 1. The 3D modeling of rotor and vertical board are first designed by using the Solidworks software. Accordingly, the 3D printer deposits the polylactic acid (PLA) layer to layer until the desired 3D object is created. After that, we coat copper foils on the rotor and vertical boards, and fix the carbon fibers on the vertical boards. Finally, a shaft with diameter of 0.7 mm is inserted into a cylindrical hole with diameter of 1.2 mm in the center of the rotor, and assembled with the vertical boards. The relevant revisions are made in **Rev 3, 14 and 17**.

Supplementary Figure 1: Fabrication process of the micromotor.

Reviewer #1: *Other forms of regenerative or low power consumption devices have been reported in the literature, though these devices are not identical to the one*

described in this paper. The paper quality may be enhanced if this device were compared to one or two other devices of similar application in terms of power density and efficiency.

Respond: Thank you very much for your suggestion. Supplementary Table 1 has summarized several low power consumption rotary micromotors, such as ultrasonic, optical and microstreaming types. Compared with these micromotors, we can find that the TM has both higher efficiency and rotate speed. It has also higher power density than the optical and microstreaming micromotors. Although the ultrasonic micromotors powered by AC power source have higher power density than the TM, the speeds are low and the driving frequencies are usually up to KHz or MHz. While the TM can be actuated by the mechanical stimuli with an ultralow frequency as low as 0.1 Hz, instead of the external high voltage power supply. Therefore, the TM has provided a novel method for a high-speed scanner actuated in ultralow frequency, which are much different and superior to previous micromotors. The relevant revisions are made in **Rev 27 and 30-31**.

Supplementary Table 1. Summary of several low power consumption rotary micromotors.

Driving type	Material	Diameter (mm)	Speed (rpm)	Power density (W/m ³)	Driving frequency (Hz)	Efficiency (%)	Ref
Ultrasonic	PZT	60	60	2420	3.9 K	—	1
Ultrasonic	PZT	1.6	430	~267000	221 K	11	2
Optical	SU-8	0.01	160	~0.3	—	~10 ⁻¹⁷	3
Optical	Liquid crystalline	0.008	660	~1.28	—	~10 ⁻¹⁸	4
Microstreaming	SU-8	0.065	625	~11.9	15 M	~10 ⁻¹⁷	5
Triboelectric	PLA	20	1300	~16	~1	41	This work

Reviewer #1: *It would have also been interesting if the authors had described more meaningfully the differences in the voltage versus sliding range and voltage versus charge loops depicted in Figure 3.d and 3.e, respectively. Why is there overlap in the plots in Figure 3.d and is there a relationship between the voltage hysteresis and the sliding range that can be described mathematically based on the ranges studied?*

Respond: Thanks for your valuable suggestions. The overlap in Fig. 3d is caused by the inappropriate range in Y-axis. We have corrected it. The voltage and charge versus sliding range are both linear.

For the *U-Q* curves versus sliding range in Fig. 3e, we have developed theoretical analysis and simulation as shown in Supplementary Note 2 and Supplementary Fig. 10. Based on the working mechanism of TM, we can find out that two electric characteristics in the rotating process. One is that the charges can be accumulated on the vertical electrodes, while the other is that the current can be generated by charges transferring from one electrode to the other. Therefore, we assume that the

micromotor can be approximately equivalent to a capacitor and a resistor in parallel, and the equivalent circuit model of the TM is depicted in Supplementary Fig. 10a. With the theoretical derivation and numerical calculation, the simulated U - Q curves versus sliding range are shown in Supplementary Fig. 10b. It is demonstrated that the output electric energy from the TENG in one cycle, described as the encircled area of U - Q curve, increases at larger sliding range, which are consistent with the experimental results in Fig. 3e. The relevant revisions are made in **Rev 11, 15, 24 and 29**.

Supplementary Figure 10: U - Q curves simulation of the TM. **a** Equivalent circuit model of the TM. **b** Simulated U - Q curves with different sliding ranges at 0.8 Hz.

Reviewer #2: *The author reported a high-speed triboelectric motor driven by sliding mode TENG. The author analyzed the effect of various parameters on the TM. The initial analysis of TEM by the author is very interesting but the application part (identity recognition) is too vague and impractical. I would like to suggest a major revision.*

Respond: Thank you very much for your time and positive comments on the manuscript. This work has realized a high-speed micromotor actuated by ultralow frequency mechanical stimuli without an external power supply, which has extended the application prospects of TENG as a high-voltage source. We have made major revisions in the application part and wish the demonstrations could validate the practicality of TM and improve the manuscript.

Reviewer #2: *The author is advised to provide the output current at different frequencies.*

Respond: We thank for your valuable advice. The relevant data have been updated in Supplementary Fig. 8. As shown, with the increase of frequency and sliding range, respectively, the measured output current peak value of the TENG with micromotor increase as well. The current peak value has approximate linear relationship with the frequency and sliding range, respectively. It is because that the sliding speed will be increased and thus the current by increasing either sliding frequency or range. Furthermore, we have measured the short-circuit current peak values of the TENG with different frequencies and sliding ranges, which are included in Supplementary Fig. 9. The relevant revisions are made in **Rev 22-23**.

Supplementary Figure 8: Measured output current of the TENG with micromotor at different frequencies and sliding ranges. **a, b** Current waveforms and peak values with different frequencies at the sliding range of 30 mm. **c, d** Current waveforms and peak values with different sliding ranges at 0.8 Hz.

Reviewer #2. *The author measured the output using Trek 347. The range of Trek 347 is 3KV but some values of output are higher than 3KV. How the author measured that high voltage.*

Respond: Thank you very much for this comment. Indeed, the actual measurable range of Trek 347 is from about -3.6 KV to +3.6 KV. We have added the $U-t$ and $Q-t$ curves of TM with different frequencies and sliding ranges as shown in Supplementary Fig. 7. The data of voltage and charge in Fig. 3 are the peak to peak values rather than peak values. We apologize for this negligent expression and have revised the Fig. 3. Just like many previous works have focused on the peak to peak voltage of TENG, such as Ref. R6 and R7, we have also selected this parameter to characterize the TM in this work. The relevant revisions are made in **Rev 9, 10 and 21**.

Supplementary Figure 7: Measured voltage and charge waveforms of the TENG with micromotor at different frequencies and sliding ranges. **a, b** Voltage and charge waveforms of the TENG with different frequencies at the sliding range of 30 mm, respectively. **c, d** Voltage and charge waveforms of the TENG with different sliding ranges at 0.8 Hz, respectively.

Reviewer #2. *The author showed open circuit voltage in the supplementary figure. The trek 347 is surface voltage measurement. The author should mention in this case how voltage varies with the load resistance.*

Respond: Thanks for your constructive suggestion. We have studied the relationship between the output voltages of the TENG on different load resistances. As depicted in Supplementary Fig. 11, with sliding range of 50 mm at the frequency of 0.8 Hz, the peak-to-peak voltage increases with the increasing load resistance until saturation. The saturation value is in accordance with the measured open-circuit voltage in Supplementary Fig. 9b. The relevant revision is made in **Rev 25**.

Supplementary Figure 11: Measured output voltage of the TENG on different load resistances.

Reviewer #2. *The author is advised to report the stability of sliding mode TENG to confirmed long-term applicability.*

Respond: Thank you very much for the reminder. The stability of sliding mode TENG are very important for the long-term applicability. As shown in Supplementary Fig. 12, the voltage waveforms have been recorded every 5000 cycles and 1000 cycles of data are presented in each recording. The voltage remains very stable after 20000 cycles, which demonstrates that the TENG has an excellent stability. The relevant revision is made in **Rev 26**.

Supplementary Figure 12: The stability of sliding mode TENG.

Reviewer #2. *The application shown by the author is simply a replication of bar code reader where scanning ray created by TM. What is the advantage of such a system over commercial barcode reader? The author should demonstrate more appropriate application for the use of TM. Furthermore, the author claimed it a self-powered system but there are so many components involves which needs external power. The self-powered word is too weak in the context.*

Respond: Thank you very much for your comments. The optical scanner is a key component in information recognition systems, which has great applications for personal digital assistant, scanner gun, intelligent gate and self-service equipment. The traditional commercial scanner usually utilizes a vibrating motor or a spin motor to form the scanning line, which has complex structure and requires a special power supply with high voltage and high driving frequency. In this work, we have first demonstrated a TM-based scanner for barcode recognition. We believe that this is not simply a replication of barcode reading. The TM-based scanner is a novel device with simple structure and easy fabrication process. More importantly, the micromotor can rotate in high speed by mechanical stimuli at ultralow frequency without external power supply. Therefore, the TM could be used for a portable scanner, in which the micromotor is actuated just by hand motion as shown in Fig. 4d and Supplementary Video 2.

Moreover, we have added a new demonstration of TM for moving obstacle detection. As shown in Fig. 4e and Supplementary Video 3, the rotational sliding TENG with tire rolling could be used for actuating the micromotor and scanning the

moving obstacle. Even though in an ultralow rotate speed, the high-speed scanning could also be realized without external power supply, which has great application prospects in vehicle-mounted laser radar and automatic drive.

As you have pointed, there are many components which needs external power in the whole systems, such as the laser and receiver. We have revised the related expressions in the manuscript. Actually, our work has focused on the power supply for the scanner by using the wasted ultralow-frequency energy in human and ambient. This is a new solution and an important progress toward a self-powered scanning identification system, which can be highly expected in the future. We wish this reply could satisfy you and receive your support. Thank you very much. The relevant revisions are made in **Rev 1, 2, 12, 13, 16, 32 and 33**.

Reviewer #2. *The author demonstrated a human identity recognition system. The reported system is very impractical for real application. The device shown in the figure looks like a contact-separation device but the author used a sliding mode device for other analysis.*

Respond: Thanks for your good question. We have removed the impractical demonstration and added a new one for moving obstacle detection, which is elaborated in Fig. 4e and Supplementary Video 3. The relevant revisions are made in **Rev 1, 2, 12, 13, 16 and 33**.

References

1. An D, Yang M, Zhuang X, Yang T, Meng F, Dong Z. Dual traveling wave rotary ultrasonic motor with single active vibrator. *Appl. Phys. Lett.*, **110**, 143507 (2017)
2. Cagaty S, Koc B, Moses P, Uchino K. A piezoelectric micromotor with a stator of $\phi=1.6$ mm and $l=4$ mm using bulk PZT. *Jpn. J. Appl. Phys.* **43** 1429–1433 (2004)
3. Kelemen L, Valkai S, Ormos P. Integrated optical motor. *Appl. Opt.* **45**, 2777–2780 (2006)
4. Ito K, Frusawa H, Kimura M. Precise switching control of liquid crystalline microgears driven by circularly polarized light. *Opt. Express* **20**, 4254–4259 (2012)
5. Kao J, Wang X, Warren J, Xu J Attinger D. A bubble-powered micro-rotor: conception, manufacturing, assembly and characterization. *J. Micromech. Microeng.* **17**, 2454–2460 (2007).
6. Li C, *et al.* Self-powered electrospinning system driven by a triboelectric nanogenerator. *ACS Nano* **11**, 10439–10445 (2017)
7. Li A, Zi Y, Guo H, Wang ZL, Fernández F. Triboelectric nanogenerators for sensitive nano-coulomb molecular mass spectrometry. *Nature Nanotech.* **12**, 481–487 (2017).

Reviewers' Comments:

Reviewer #1:

Remarks to the Author:

I have reviewed both my comments/suggestions and those of the other reviewers and am very satisfied with the modified document. The overall quality of the paper has improved with significant modifications and inclusion of supporting evidence data and videos supporting the work. The work is both novel and also demonstrative of growth in this area. I have no additional requests for modification to the draft.

Reviewer #2:

Remarks to the Author:

Triboelectric micromotors actuated by ultralow frequency mechanical stimuli

I want to suggest minor revision to address the following comment.

1. The author measured the load resistance. How the load resistance was measured using surface voltage unit trek 347. The author should include the circuit and detailed explanation regarding the measurement.

Revisions and Responds to Reviewers

Manuscript ID: NCOMMS-18-30792A

Manuscript Type: Article

Title: Triboelectric micromotors actuated by ultralow frequency mechanical stimuli

Author(s): Hang Yang, Yaokun Pang, Tianzhao Bu, Wenbo Liu, Jianjun Luo, Dongdong Jiang, Chi Zhang, Zhong Lin Wang

Corresponding author: Chi Zhang, czhang@binn.cas.cn; Zhong Lin Wang, zlwang@gatech.edu

List of revisions made to the manuscript and supplementary information:

Rev 1. Page 9, line 5-6: The diagrammatic illustration is shown in Supplementary Figure 13.

Rev 2. Supporting Information, Page 7, line 7-9: Supplementary Figure 13.

Respond to the reviewers:

Reviewer #1: *I have reviewed both my comments/suggestions and those of the other reviewers and am very satisfied with the modified document. The overall quality of the paper has improved with significant modifications and inclusion of supporting evidence data and videos supporting the work. The work is both novel and also demonstrative of growth in this area. I have no additional requests for modification to the draft.*

Respond: Thank you very much for your positive comments.

Reviewer #2: *I want to suggest minor revision to address the following comment.*
1. The author measured the load resistance. How the load resistance was measured using surface voltage unit trek 347. The author should include the circuit and detailed explanation regarding the measurement.

Respond: Thank you very much for your suggestion. During the measurement, one end of the measured device was grounded, while the other end was attached to a copper film which was 5 mm beneath the probe. We have included the circuit illustration to explain the measurement in Methods and Supplementary Figure 13. The relevant revisions are made in **Rev 1 and 2**.